# Preparing for Patient-Customized N-of-1 Antisense Oligonucleotide Therapy to Treat Rare Diseases

**DOI:** 10.3390/genes15070821

**Published:** 2024-06-21

**Authors:** Harry Wilton-Clark, Eric Yan, Toshifumi Yokota

**Affiliations:** 1Department of Medical Genetics, Faculty of Medicine and Dentistry, University of Alberta, Edmonton, AB T6G 2R3, Canada; hwiltonc@ualberta.ca; 2Department of Biological Sciences, Faculty of Science, University of Alberta, Edmonton, AB T6G 2R3, Canada; eyyan@ualberta.ca

**Keywords:** antisense oligonucleotide, N-of-1, milasen, atipeksen, valeriasen, rare disease, exon skipping, splice switching, personalized medicine, muscular dystrophy

## Abstract

The process of developing therapies to treat rare diseases is fraught with financial, regulatory, and logistical challenges that have limited our ability to build effective treatments. Recently, a novel type of therapy called antisense therapy has shown immense potential for the treatment of rare diseases, particularly through single-patient N-of-1 trials. Several N-of-1 antisense therapies have been developed recently for rare diseases, including the landmark study of milasen. In response to the success of N-of-1 antisense therapy, the Food and Drug Administration (FDA) has developed unique guidelines specifically for the development of antisense therapy to treat N-of-1 rare diseases. This policy change establishes a strong foundation for future therapy development and addresses some of the major limitations that previously hindered the development of therapies for rare diseases.

## 1. Introduction

### 1.1. The Burden of Rare Diseases

As our understanding of medicine and precision diagnostics grows, the burden of rare diseases continues to become more and more evident [1]. While the exact definition of a rare disease varies by locale—defined as affecting fewer than 200,000 Americans in the USA or <1/2000 individuals in Europe and Canada—the global toll that these diseases take is undeniable [1,2]. Although the prevalence of each disease individually is low, the summed impact of all rare diseases becomes salient at a population level, and it is estimated that between 3.5% and 5.9% of the global population has a rare disease [3].

Despite our growing ability to diagnose rare diseases, developing treatments for these conditions is hampered by a plethora of difficulties [4,5]. Due to their low individual prevalence, minimal data are available in databases, making it more difficult to elucidate their varied mechanisms and design appropriate therapies. This is even more pertinent in pediatric disorders—estimated to be 50–75% of all rare diseases—where the progressive or fatal nature of many diseases severely limits our ability to study and understand them [3,6]. The lack of data is further compounded by the complex diagnostic odyssey that is often required for these patients, as many clinicians will have encountered very few similar patients. Funding for the development of rare disease therapies is also a major hurdle [7]. Given that only a single patient is originally treated with these therapies, financing the high pharmaceutical development costs that these studies entail is extremely difficult as commercial interest is low. For standard drug development, the median cost of development to receive FDA approval is estimated at $985.3 million USD [8].

Finally, region-specific regulatory requirements such as clinical trials through the FDA are difficult to complete. Due to the low prevalence and pediatric onset of many rare diseases, finding enough patients to satisfy regulatory mandates can be near impossible for some populations [9]. It has been suggested that the typical randomized control trials, viewed as the gold standard in drug development, could be reconsidered or loosened in the context of rare disease to enable therapy development [2]. Due to the difficulties with standard therapy development in the context of rare diseases, some groups have instead opted for an N-of-1 approach, where a novel therapy is custom-designed for a single patient who has no access to other treatment options [10]. While this approach faces many of the same financial and regulatory limitations as other rare disease therapies, it bypasses the need for a large patient population and instead develops the therapy based on a single patient partner.

### 1.2. Antisense Therapy

In the last several years, a treatment approach known as antisense therapy has gained popularity as a possible treatment for rare diseases, particularly in the context of N-of-1 therapy development [11,12,13]. Antisense therapy uses synthetic ribonucleic acid (RNA)-like oligonucleotides known as antisense oligonucleotides (ASOs) to treat disease by modulating the protein expression of a target gene [14]. These ASOs are synthetic molecules designed to mimic the structure and function of natural RNA, and are engineered to enhance stability, binding affinity, and specificity for their target RNA sequences. They are often used in various therapeutic and research applications to modulate gene expression or correct genetic mutations. ASO-mediated modulation can occur either through transcript degradation, translational regulation, or splice-switching (Figure 1) [15,16]. For transcript degradation, ASOs with a sequence complementary to the transcript of interest are designed using base pair and backbone chemistries that are susceptible to ribonuclease H. Upon binding to the target transcript through Watson–Crick base pairing, the RNA–ASO duplex is degraded by ribonuclease, leading to knockdown of the target gene (Figure 1A) [17]. This method is ideal when the disease arises from the presence of a pathogenic transcript, such as in toxic gain-of-function mutations. For splice-switching, ribonuclease-resistant ASOs are instead designed to bind to important splice sequences such as acceptors, donors, silencers, or enhancers. Rather than stimulating degradation, these ASOs sterically hinder spliceosome binding, leading to the inclusion or exclusion of a given intron or exon (Figure 1B) [16]. A common type of splice-switching is exon skipping, such as in the context of Duchenne muscular dystrophy (DMD), where the exclusion of mutated exons to restore the reading frame can confer clinical benefit [18]. Rather than altering splice patterns, ribonuclease-resistant ASOs can also be targeted to ribosomal binding sites to downregulate expression by impairing ribosomal binding (Figure 1C) [18].

Although ASOs have been used for therapeutic purposes since 1998, their popularity has seen a sharp rise in the last decade [19]. Fomiversen, an antiviral, was the first ASO to receive FDA approval for the treatment of cytomegalovirus retinitis in 1998 [20]. Following a 15-year lull, mipomersen also received FDA approval in 2013 for the treatment of familial hypercholesterolemia [21,22]. Since then, the rate and number of ASOs under development have increased dramatically, and numerous ASOs have received FDA approval in the last decade: eteplirsen (DMD—2016), nusinersen (spinal muscular atrophy—2016), inotersen (hereditary transthyretin-mediated amyloidosis—2018), golodirsen (DMD—2019), viltolarsen (DMD—2020), casimersen (DMD—2021), and tofersen (amyotrophic lateral sclerosis—2023) [23,24,25,26,27,28,29]. The relatively high volume of ASOs for DMD versus other indications stems from the fact that ASOs for DMD use the exon skipping approach, removing specific exons to restore the normal function of proteins. This means that theoretically, different ASOs could be designed for each of the 79 exons in *DMD*, leading to multiple different ASOs for different mutations causing DMD, as opposed to knockdown approaches where a single ASO can knock down the entire gene. The popularity of ASOs for all indications is only expected to increase with time, and a review from early 2022 reported that at that time, there were 80 different ASOs in either phase II or phase III clinical trials for myriad different diseases [30].

As previously established, the development of N-of-1 and rare disease therapies has historically been constrained by numerous challenges, including regulatory and financial hurdles. ASOs harbor numerous properties that make them well suited for addressing these issues. They have a well-established safety profile, and due to their popularity, the required safety screening and possible complications are well understood [31]. Furthermore, the sequence-specific binding of ASOs allows for extreme precision in N-of-1 approaches, where the oligonucleotide (oligo) can be targeted to a patient’s exact mutant sequence. Given their relatively simple design, ASOs are also relatively cheap to develop compared to other biologics, making them an attractive choice for N-of-1 approaches that are not expected to generate significant external investment [32]. The Boston Children’s Hospital, where two different N-of-1 ASOs have been developed, stated that the development of these approaches costs an average of USD 1.6 million [33]. While this is still a prohibitively high cost for many patients that requires substantial fundraising efforts, it represents a fraction of the USD 985.3 million average cost of development for FDA-approved drugs [8]. Based on these factors, ASOs have begun to be explored as unique candidates for developing N-of-1 therapies for rare diseases. Here, we aim to provide an overview of the cases of N-of-1 ASO development to date, as well as outline how the FDA regulatory requirements have shifted to enable this cutting-edge approach to be better used to treat rare diseases.

## 2. Cases of N-of-1 ASO Development

### 2.1. Milasen: N-of-1 Exon Skipping to Treat Batten Disease

The development of milasen was a pivotal milestone in the popularization of N-of-1 antisense therapies [34]. As one of the first oligos tested in this capacity, milasen demonstrated the feasibility of this approach and substantially contributed to the creation of the current FDA guidelines for N-of-1 antisense therapy trials.

The development of milasen began in 2017 after a then-6-year-old girl, Mila, was diagnosed with late-infantile Batten disease, a fatal neurological disorder belonging to a category of diseases known as neuronal ceroid lipofuscinoses (NCLs), which affect approximately 1/100,000 people worldwide [34,35,36]. NCLs are a multigenic set of neurodegenerative diseases caused by the accumulation of toxic lipofuscinin neurons and organs, typically characterized by early-onset seizures, visual impairments, and developmental delay. Patients typically face severe loss of vision and speech by their early teens, and few patients survive beyond their late teen years [36].

Following the rapid onset of vision loss and other hallmark symptoms at age 6, Mila was referred to genetic testing, where she was confirmed to harbor mutations in both alleles of *CLN7*, also known as *MFSD8*, one of the genes associated with late-infantile-onset Batten disease. One allele contained a known pathogenic variant, while the other contained a novel 2 kb insertion in intron 6 that was modulating splicing to promote the inclusion of intron 6 in mature mRNA, leading to a dysfunctional protein product [34]. Based on this latter mutation, it was theorized that ASOs targeting the cryptic intron 6 splice site may be able to prevent its inclusion, restoring appropriate CLN7 expression to treat Mila’s disease.

Splice-switching 2-methoxyethyl (2′-MOE) and 2′-Ome ASOs were designed using in silico predictions and tested both in vitro and in vivo to confirm their efficacy and safety. ASOs were provided to patient fibroblasts, and efficacy was assessed using qRT-PCR to measure the proportion of healthy *CLN7* transcripts [34]. Preliminary efficacy was also gauged via in vitro markers of disease progression, such as intracellular vacuolization and lysosomal mass. Toxicity was assessed by monitoring rats injected with 2.5-fold, 10-fold, and 42-fold the expected dose over the course of 70 days. Notably, the predicted dose was selected based on the human dose of nusinersen, an FDA-approved ASO with similar chemistry used to treat spinal muscular atrophy [25,34]. Rats in the high-dose cohort displayed dorsal root ganglion toxicity and gait disturbance, which was used to inform toxicity monitoring during the clinical phase of the study.

Given Mila’s rapid deterioration and lack of suitable treatment alternatives, clinical investigational treatment was started shortly after concluding the safety studies. Mila was treated with bi-weekly intrathecal injections of milasen for 4 months, followed by quarterly maintenance doses [34]. There were no serious adverse effects, and electroencephalography found that seizure frequency and duration were reduced to half of their pretreatment values. Unfortunately, milasen failed to impede progressive brain volume loss, and Mila passed away three years after beginning treatment [34].

The story of milasen became a landmark study for both precision medicine and Batten disease. The total development of milasen took shortly over one year, an incredible achievement that showcases the potential of ASOs for rapid mobilization and N-of-1 approaches. As the first case of N-of-1 ASO development, it also led to the creation of substantial regulatory and financial infrastructure that opened the doors for future N-of-1 studies, such as the FDA N-of-1 guidelines and multiple not-for-profit organizations like the N = 1 Collaborative [33]. It also inspired other studies exploring the use of ASOs to treat CLN3 juvenile Batten disease, and preclinical studies using ASOs to treat *CLN3* mutations are already underway [37,38].

### 2.2. Atipeksen

Following the precedent set by Milasen, Kim et al. developed a personalized ASO to treat an individual with ataxia–telangiectasia (A-T), a rare neurological syndrome affecting an estimated 1/40,000 to 1/100,000 children worldwide [39]. A-T, also called Louis–Bar syndrome, is caused by biallelic mutations in the *ATM* gene on human chromosome 11q22.3 [40]. The affected protein, ATM, is a serine/threonine kinase from the phosphoinositide 3-kinase-related kinase family with functions in cell cycle checkpoint signaling and DNA damage response.

A-T shows variable expressivity, with clinical presentation and rate of disease progression differing case by case; however, the syndrome is generally characterized by cerebellar degeneration and ataxia or decreased coordination of movements during school years [41]. This often manifests as decreased balance while sitting, standing, and walking, as well as issues with fine motor function, eye movement, and speaking. In the classic form of A-T, the first symptom observed is typically ataxic gait, with a median onset of 1.5 years [42]. Additional symptoms can include telangiectasia, immune deficiencies, impaired lung function, increased susceptibility to cancer and diabetes, and growth delays due to hormonal abnormalities. A-T is managed by treating the symptoms that manifest in a given patient [43]. Notably, recurrent lung infections remain a serious complication, and patients with A-T are recommended to take regular lung function testing. Cancer screening is also an important management tool, and frequent screening is recommended, beginning in early childhood, although no definitive guideline exists for cancer surveillance in A-T [44]. More recently, drugs to slow disease progression are currently being explored in clinical trials. Nicotinamide riboside and NAD supplementation is a promising candidate treatment, where supplementation has been associated with improved performance on neurological and motor tests [45].

Following the development of milasen by the same group, Kim et al. aimed to create guidelines for the rational development of N-of-1 antisense oligonucleotide therapy for genetic diseases and implement it to generate an N-of-1 ASO to treat A-T [39]. First, whole-genome sequencing was completed on A-T patient samples, identifying disease-causing variants: 75% of variants were thought to cause loss of function in *ATM*, and 16% were thought to cause loss of function specifically by destroying splice sites. The team then developed an in silico algorithm to evaluate whether *ATM* variants could be amended by splice-switching ASO. Following this criterion, 15% of the patient samples had variants predicted to be possibly or probably ASO-amenable, the majority of which were caused by mutations in intronic sites.

The authors then developed a splice-switching ASO treatment for a single pathogenic variant as an N-of-1 treatment. The female patient, Ipek, was 1 year of age at referral and carried compound heterozygous loss-of-function mutations in *ATM*: a 13 bp deletion (c.8585-13_8598del) in one allele and a c.7865C>T point mutation in the other. The latter is predicted to produce a novel splice donor site within exon 53 and cause protein frameshift by excluding 64 bp of the exon in the mature mRNA. Previous work has shown that ASO was able to effectively block the novel splice site and restore ATM protein function in a c.7865C>T cell line, providing precedence for N-of-1 ASO treatment for the c.7865C>T allele [46]. Kim et al. designed 32 total candidate ASOs targeting the novel splice donor site or adjacent splice silencer sites using phosphorothioate 2^-O-methoxyethyl backbone chemistry. ASO efficacy was evaluated in patient fibroblasts. Cells were transfected with 200 nM of ASO, RNA was extracted 24 h post-transfection, and RT-PCR was completed to identify proportions of normal *ATM* splicing. Of note, the strongest candidate was AT008, which encompassed the novel splice site and restored up to 50% normal *ATM* splicing, though it also induced exon 53 skipping. qRT-PCR was later completed, which identified that AT008 and another strong candidate, AT026, produced 29% and 18% functional *ATM* transcript, respectively. Several candidates were validated by assessing protein function. The authors assayed one of the normal functions of ATM, phosphorylation of p53 and KAP1 upon radiation exposure, by an immunoblot assay. ASO treatment caused restoration of p53 and KAP1 phosphorylation compared to untreated controls as well as a hypomorphic variant, showing strong therapeutic promise. For safety analysis, AT026 and AT008 were aligned in silico to the human genome, where off-target effects were predicted to be minimal. Also, in vitro toxicity was assessed with an annexin V and propidium iodide apoptosis assay, where AT008 showed similar apoptosis profiles to random-sequence ASO, acting as evidence of a lack of sequence-specific functional effects. Safety was assessed in vivo following the same methods as during the development of milasen.

At this point, ASO candidate AT008 was renamed atipeksen and chosen to be an N-of-1 treatment for Ipek’s disease. ASO administration began at 2 years and 9 months of age in early 2020, where Ipek was given atipeksen injections every 2 weeks, escalating the drug dose from 3.5 mg to 42 mg over 10 weeks. Intrathecal injection was chosen rather than intracerebroventricular due to it being less invasive and carrying a lower operative risk. Maintenance doses of 42 mg were then administered every 8 weeks, after which the dose was adjusted to 63 mg. Throughout the atipeksen administration, there were several interruptions to dosing. Ultimately, a maintenance dose of 42 mg was given every 12 weeks. During this period, Ipek’s safety was monitored via blood and cerebrospinal fluid biomarkers and her disease progression was evaluated with neurological and physical examinations as well as blood and cerebrospinal fluid biomarkers. At the time of publication, Ipek was aged 6 and had been treated for 3 years, and the authors reported Ipek’s preliminary clinical examination scores to be mild for her cohort of A-T patients [34,47]. The case of atipeksen serves as a proof of concept for the proposed guide to N-of-1 ASO development, but also importantly was a landmark genetic treatment for another disease lacking causative therapies.

In addition to milasen and atipeksen, several other N-of-1 ASOs have been developed that to the best of the authors’ knowledge have no associated academic or official publications available. While not as thorough as the process for atipeksen or milasen, some information is available regarding these cases from patient-run sites and news articles. Given that these are not peer-reviewed sources, diligence must be exercised when drawing any conclusions from these sources.

### 2.3. Valeriasen

In 2018, a girl, Valeria, presented with a seizure shortly after birth and was diagnosed with epilepsy arising from a c.1421A>G mutation in *KCNT1*, which encodes for a sodium-activated potassium channel [46,48]. Gain-of-function mutations in *KCNT1* like these are associated with increasing conductance to potassium and inhibition of inhibitory neurons, causing epilepsy. Previous work unrelated to Valeria had generated gapmer ASOs that were effective at knocking down pathogenic *KCNT1* via RNase H degradation of transcript in vivo in a mouse model [49]. ASO administration in symptomatic mice reduced seizure frequency and behavioral abnormalities while increasing survival. The team treating Valeria developed a similar ASO, valeriasen, which suppressed her pathogenic KCNT1 in vitro and completed animal safety studies with ASO over the course of 8–10 weeks. Valeriasen was administered intrathecally in a dose-escalation period starting in 2020. Unfortunately, Valeria lost her life 12 months later due to hydrocephalus [50]. The results of the investigation regarding the cause of the hydrocephalus have not been made available; however, hydrocephalus has also been reported in patients treated with intrathecally injected ASO for Huntington’s disease and spinal muscular atrophy [51,52]. Alternatively, hydrocephalus and epilepsy are often symptoms of a single underlying neurological cause [53]. Currently, the foundation established by Valeria’s parents is developing a revised version of valeriasen to treat *KCNT1* mutations and working on improving safety screening for ASO therapies.

### 2.4. Locked Nucleic Acids (LNAs) for TNPO2-Related Epilepsy and Developmental Delay

In 2021, Leo, a newborn boy, showed symptoms of microcephaly and seizures, leading to a diagnosis of developmental delay and epilepsy stemming from a rare *TNPO2* single-nucleotide variant causing toxic gain of function [53,54]. Although only a small quantity of data are available for Leo’s journey, it appears that his parents collaborated with a biotechnology company, Creyon Bio, to design gapmer ASOs with LNA chemistry to knock down the toxic allele while not affecting the healthy allele. Several candidates showed efficacy in vitro and fulfilled in vitro and in vivo safety criteria. Patient administration began in July 2023 via intrathecal injection with a dose escalation to 40 mg [53]. Cerebrospinal fluid markers and MRI results indicate the ASO was well tolerated while seizure frequency was reduced and previously regressed developmental milestones were regained, clearly showing ASO caused clinical improvement.

### 2.5. Antisense Therapy for KIF1A-Associated Neurological Disorder (KAND)

In 2016, Susannah, a girl of 2 years of age, was diagnosed with KIF1A-associated neurological disorder caused by a mutation in the *KIF1A* gene [55]. This disease has highly varied manifestations, showing both autosomal-dominant and -recessive inheritance patterns. Phenotypes also vary: patients most commonly show seizures but developmental delays, cerebellar degeneration, and loss of sensation are present in different forms of the disease [56]. Susannah’s family worked with a nonprofit organization, n-Lorem, as well as Ionis Pharmaceuticals, to design and manufacture an ASO treatment for her specific mutation, which was administered to the spine [57]. Immediately following her second dose of ASO, Susannah’s father reported that her speech and vision issues seemed to have improved. No further details regarding ASO type or binding location are available to the best of the authors’ knowledge.

## 3. Developing N-of-1 Therapies: FDA Regulatory Guidelines

Regulatory constraints are an important aspect of development of N-of-1 therapies for rare diseases [9]. Factors such as low prevalence, rapidly progressive diseases, and poor patient registries can make it nearly impossible to generate the high-volume data that are typically required for new drug development. Thus, a streamlined approach that permits the development of therapies for rare diseases was needed to help these patients. Based on the success of milasen and atipeksen and the well-characterized safety profile of ASOs, the FDA developed novel guidelines specifically for the development of ASO therapies for N-of-1 indications [58]. N-of-1 ASO development is intended to be used when there are no other FDA-approved options available for a patient and the disease being treated shows rapid progression leading to death or serious impairment. The guidelines below outline the areas that must be addressed in investigational new drug (IND) applications, with major components including a non-clinical report, chemistry and manufacturing report, and clinical plan.

### 3.1. Non-Clinical Report

Given the unique genetic targeting inherent to an N-of-1 ASO approach, these therapies are extremely personalized. Due to their mutation-specific nature and the short turnaround time often required for their development, testing N-of-1 ASOs in humans is not feasible. Thus, the animal model work in the non-clinical report comprises the majority of the efficacy and safety studies in the IND [59]. The guidelines suggest that a single toxicity study that adequately assesses pharmacological safety in the cardiovascular, central nervous, and respiratory systems is sufficient to support an IND. This study can be completed in a rodent or non-rodent model and should be 3 months in duration; however, in the case of a disease expected to progress to significant morbidity within one year, the IND can be submitted after two weeks of in-life toxicity data are generated. Such was the case with milasen, where safety testing was conducted through an accelerated 70-day trial in rats [34]. It is recommended that the tested dose is close to the maximally tolerated dose in the chosen model. Off-target effects must be assessed in silico with the Basic Local Alignment Search Tool (BLAST), but otherwise no separate experiments are required to assess for off-target effects or genotoxicity [59].

This 3-month single test represents a substantial acceleration of the usual preclinical safety testing performed during standard clinical trials. Comparatively, the preclinical testing for casimersen, a recently approved oligonucleotide treating Duchenne muscular dystrophy, included three different animal models, studies spanning nearly one year, and discrete testing for genotoxicity and reproductive toxicity (Table 1) [26,60]. The rapid nature of safety testing is required in order to treat these N-of-1 patients as urgently as possible and is made more feasible by the previously established safety profile of ASOs in other trials. However, the lessened testing may come with additional toxicity risk. This increased risk is important to disclose and must be carefully weighed against the consequences of untreated disease progression.

In addition to safety, the non-clinical report must also assess the efficacy of the investigational ASO. While the specific assays and endpoints will vary for each disease, a general rubric is provided [59] For any in vitro or in vivo efficacy assay performed, a report should be provided stating the purpose of the study, a detailed description of the study design, and all data and analyses generated from the study.

### 3.2. Chemistry, Manufacturing, Control (CMC) Report

The guidelines also outline the chemistry and manufacturing information that must be submitted in the IND. First and foremost, the N-of-1 ASO pathway is intended for ASOs from a well-characterized chemistry, such as 2′-MOE-substituted oligonucleotides or phosphorodiamidate morpholino oligomers (PMO) [61]. Notably, these guidelines also exclude conjugated ASOs. ASO–peptide conjugates have generated considerable interest recently due to their improved tissue targeting and cardiac efficacy, and the most advanced ASO–peptide (SRP-5051) is currently in phase III clinical trials for the treatment of DMD [62,63].

The CMC report should include detailed information regarding the drug substance, such as structure, nomenclature, structural and molecular formulas, molecular weight, physical properties, and molecular weight of any salts or excipients. Particular focus should be given to the chemistry of the oligonucleotide base and backbone [61]. These properties should be further confirmed through techniques such as nucleotide sequencing and mass spectrometry. Details should also be provided outlining the detailed manufacturing and sterilization process, including all components required, as well as any potential impurities that can arise during the synthesis process. Finally, the CMC report should include control measures and specifications to which each batch of ASO should conform, including the specific tests, acceptance criteria, and appropriate storage conditions [61].

### 3.3. Clinical Considerations Report

The clinical report guidelines take a more prospective view than the non-clinical and CMC reports. It outlines the ethical and administrative requirements for proceeding with in-human testing, as well as a detailed overview of the safety and clinical efficacy monitoring that will be performed once in-human dosing is commenced [64]. Firstly, the guide states that the protocol for ASO administration must be approved by the institutional review board (IRB) and that informed consent must be obtained from the patient or the patient’s legally authorized representative. During these disclosures, particular attention should be drawn to the experimental first-in-human nature of the ASO and whether the risks of this approach are justified by the anticipated clinical benefit of treatment. Given the unique circumstances of these IND applications, it is also recommended that the sponsors consider discussing their proposal with a professional medical ethicist prior to filing their application. The clinical report should also include the rationale for the treatment approach, including confirmation of the patient’s genetic diagnosis, evidence supporting the role of the targeted gene in disease pathogenesis, and evidence that the identified variant is unique to the patient.

To ensure safety and consistency, it is recommended that the same batch of ASOs used in the non-clinical studies supporting the IND should also be used for clinical investigations where possible [64]. Initial dose should be calculated using a human-equivalent dose conversion based on the animal testing conducted in the non-clinical studies, and should be supported by the existing data from previously approved ASOs of the same chemical class [65]. The method of administration, including method and frequency, should also be identified at this stage and informed by previous testing.

A detailed overview of clinical safety procedures should be included in the clinical report. Given the truncated non-clinical safety testing that is required for the IND application compared to standard clinical trials, rigorous monitoring during the clinical study is of the utmost importance to ensure the safety of IND participants. A detailed outline of planned safety assessments and schedule should be included, monitoring for adverse events and toxicology based on the specific ASO used and risks previously identified in the non-clinical work. Safety assessments should be performed before each dose at a minimum, with a higher frequency in the early months of the study. When phosphorothioate backbones are included in the ASO, care should be taken to monitor for thrombocytopenia, which has been a major concern in previous trials [66].

Lastly, the assessment of clinical response throughout the treatment period should be highlighted. Case-specific metrics of disease progression should be identified in advance and assessed at predefined intervals to determine overall clinical benefit. These metrics will also be used to continually reassess the clinical response-to-risk ratio, ensuring that continuation of the IND remains in the best interest of the patient. The exact metrics assessed will vary from disease to disease, and can include clinical scales, performance metrics, caregiver-reported outcomes, or scientifically relevant biomarkers. Longer-term, annual safety reports should be generated summarizing adverse effects and efficacy over the preceding year.

## 4. Conclusions

The effective development of therapies for rare diseases through conventional methods is severely limited by financial, logistical, and regulatory hurdles. By using antisense oligonucleotides in an N-of-1 development approach, many of the financial and logistical hurdles can be bypassed, offering a unique approach to rare disease treatment. Furthermore, based on several recent N-of-1 ASOs, the FDA has created an independent trial pipeline specifically for N-of-1 ASOs that alleviates some of the regulatory hurdles associated with development. As a result, ASOs are now uniquely positioned to be used as N-of-1 therapeutic agents in a capacity that has not previously been possible, laying the groundwork for notable advances in the future with regard to the treatment of rare diseases.

## Figures and Tables

**Figure 1 genes-15-00821-f001:**
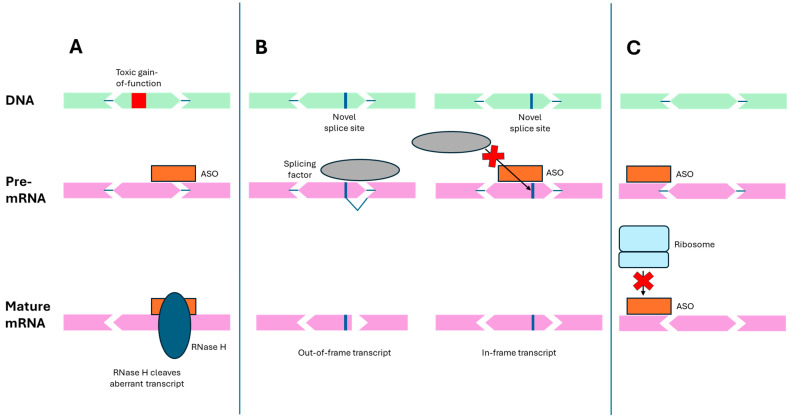
Common mechanisms of antisense oligonucleotides. **A.** RNase H-mediated degradation. ASO–mRNA double-stranded complex recruits RNase H. **B.** Correction of aberrant splicing. ASO binds splice sites on the transcript and sterically blocks binding of splicing factors. This approach can also be used to prevent erroneous splicing, as pictured, or can be used to exclude targeted exons. **C.** Reduction in translation. ASO sterically blocks ribosome binding to the transcript.

**Table 1 genes-15-00821-t001:** Summary of non-clinical safety studies in the FDA guidelines compared to milasen and casimersen.

	ASO Guidelines	Milasen	Casimersen
Duration prior to clinical studies	8–12 weeks	4 weeks	39 weeks
Number of animal models tested	Single	Single	Multiple
Species	Rodent or non-rodent	Female rats	Male mice, male rats, male monkeys
Genotoxicity	Not required	Not assessed	Chinese hamster ovary (CHO) chromosomal aberration assay, mouse bone marrow micronucleus assay
Reproductive toxicity	Not mentioned	Not assessed	Analysis in monkeys of sperm count, motility, and morphology, monkey hormone level analysis
Off-target analysis	In silico BLAST	In silico BLAST	In silico BLAST

## Data Availability

No new data were created or analyzed in this study. Data sharing is not applicable to this article.

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
