# Peer review of "Preparing for Patient-Customized N-of-1 Antisense Oligonucleotide Therapy to Treat Rare Diseases"

_genes, 2024, doi:10.3390/genes15070821_

Round 1

Reviewer 1 Report

Comments and Suggestions for Authors

The authors provide a timely, albeit brief, review of the changing landscape of therapeutic options for patients with rare diseases. The authors predominantly focus on antisense technology while discussing the changing FDA regulatory landscape for novel treatments for rare diseases. However, the authors do mention some other options. The focus on antisense allows clear discussion of specific ways the technology can be employed, specific case studies of regulatory approval, and specific case studies of efficacy. Additionally, the authors provide a nice summary table of the steps involved in advancing an ASO for a patient. In sum, this short review is timely, accurate, and should be an excellent fit for the special issue.

Author Response

Reviewer 1: (Comments underlined, responses in blue)

The authors provide a timely, albeit brief, review of the changing landscape of therapeutic options for patients with rare diseases. The authors predominantly focus on antisense technology while discussing the changing FDA regulatory landscape for novel treatments for rare diseases. However, the authors do mention some other options. The focus on antisense allows clear discussion of specific ways the technology can be employed, specific case studies of regulatory approval, and specific case studies of efficacy. Additionally, the authors provide a nice summary table of the steps involved in advancing an ASO for a patient. In sum, this short review is timely, accurate, and should be an excellent fit for the special issue.

Thank you for taking the time to review our manuscript. Your comments and thoughts are sincerely appreciated.

Reviewer 2 Report

Comments and Suggestions for Authors

These authors describe the on-going strategy and the current situation of N-of-1 ASO therapy especially for rare diseases. This review is well written and concise. For readers who are not so familiar with ASO therapy, this review is a milestone to know the overview of this field and also to summarize this field for the specialist. Some issues described below should be considered and revised.

In the Introduction, examples of ASOs that received FDA approval are presented. Most of these ASOs are for DMD, but it would be better to briefly include the authors' opinions on the history of this reason why ASOs are for DMD a lot. The opinions on why ASOs for SOD1-ALS are only one would be better to be included. Furthermore, it would be feasible and easy to read if there is a Table summarizing the indications for N-of-1 ASO therapy, their ASO sequences (if they are publicly available), targets/names, etc.

In line 64, they use “RNA-like oligonucleotides”. Please clarify what RNA-like is. Moreover, please specify the difference between LNA (in Section 2.4) and RNA-like.

In Section 3.1, it is good to show a description of non-clinical reports. However, since only guidelines are given here, it would be better to add some (typically, important, or famous) specific examples of ASOs that have been applied to animal models.

Author Response

Reviewer 2: (Comments underlined, responses in blue)

These authors describe the on-going strategy and the current situation of N-of-1 ASO therapy especially for rare diseases. This review is well written and concise. For readers who are not so familiar with ASO therapy, this review is a milestone to know the overview of this field and also to summarize this field for the specialist. Some issues described below should be considered and revised.

Thank you for taking the time to review our manuscript. Your comments and thoughts are sincerely appreciated.

In the Introduction, examples of ASOs that received FDA approval are presented. Most of these ASOs are for DMD, but it would be better to briefly include the authors' opinions on the history of this reason why ASOs are for DMD a lot. The opinions on why ASOs for SOD1-ALS are only one would be better to be included. Furthermore, it would be feasible and easy to read if there is a Table summarizing the indications for N-of-1 ASO therapy, their ASO sequences (if they are publicly available), targets/names, etc.

A brief explanation for the high number of DMD-specific ASOs has been included, and now reads: “The relatively high volume of ASOs for DMD versus other indications stems from the fact that ASOs for DMD use the exon skipping approach, removing specific exons to restore the normal function of proteins. This means that theoretically, different ASOs could be de-signed for each of the 79 exons in DMD, leading to multiple different ASOs for different mutations causing DMD, as opposed to knockdown approaches where a single ASO can knock down the entire gene.”. The authors originally planned to include a table very similar to the one outlined, however we elected to exclude it given the low amount of specific information available for many of the N-of-1 ASOs that we discuss in the manuscript, which led to a very incomplete table.

In line 64, they use “RNA-like oligonucleotides”. Please clarify what RNA-like is. Moreover, please specify the difference between LNA (in Section 2.4) and RNA-like.

The RNA abbreviation has been spelled out, and distinguished from LNA which is a specific chemistry of ASO. The sections now read “Antisense therapy uses synthetic ribonucleic acid (RNA)-like oligonucleotides known as antisense oligonucleotides (ASOs) to treat disease by modulating protein expression of a target gene [14][14]. These ASOs are synthetic molecules designed to mimic the structure and function of natural RNA, and are engineered to enhance stability, binding affinity, and specificity for their target RNA sequences. They are often used in various therapeutic and research applications to modulate gene expression or correct genetic mutations.” and “ …to design gapmer ASOs with LNA chemistry to knock…”.

In Section 3.1, it is good to show a description of non-clinical reports. However, since only guidelines are given here, it would be better to add some (typically, important, or famous) specific examples of ASOs that have been applied to animal models.

Important examples of animal model testing for N-of-1 ASOs are outlined in Section 2 within the subsection for each respective ASO. The clarity of this section has been improved by alluding to section 2. It now reads: “Such was the case with Milasen, where safety testing was conducted through an accelerated 70-day trial in rats [34]”

Reviewer 3 Report

Comments and Suggestions for Authors

Genes – 3056569 : Preparing for N-of-1 ASO treatments for rare diseases, Clark, Yan & Yokota  Comments for authors, 11.06.24.

(The attached pdf is a copy of this report, to ensure formatting of the Table below is preserved)

This is a very readable and fascinating paper for general readers, in which the authors point to the future by compiling a current summary of published examples of single-patient antisense oligonucleotide (ASO) treatments for individual rare genetic disorders. The authors also present some of the background considerations regarding the statutory regulation in the US for the development of such treatments, albeit at a quite superficial level, but which is probably inevitable in a summary paper.  The narrative style of the case histories, including referring to each patient by name, tends to a flowing journalistic style, rather than the more customary succinct scientific style of writing, and although to some the narrative style could be irritating, it should not necessarily require revision.   However, in relation to this there is a Major point that does need attention:

Major Point

1.Given that each patient’s name is already public, and that the name given to each patient-specific ASO is based on the patient’s name, the use of the patient’s name is understandable.  However, the authors do need to add a note in the Acknowledgements and Informed Consent Statement about this, and confirm that the parents of each child has given a general permission for the child’s name to be used in any publication which includes the details of their case.  Accordingly, I would question whether the Informed Consent Statement as ‘not applicable’ is an acceptable response by the authors.        

2. The three example-types of ASO in the diagrams in Figure 1, do not make any reference to use of ASOs to give ‘exon-skipping’ as an approach to re-correction of a reading frame in cases where there is an out-of-frame exon deletion.  Since this is the most common mutation mechanism in Duchenne muscular dystrophy, the authors need to comment on this, and, if appropriate, either add a fourth mechanism to Figure 1, or mention in the text and legend if it is already covered by one of the three mechanisms present. It maybe, that given the high incidence of DMD, and the number of patients with similar mutations, the authors may not consider this to be within an ‘N-of-1’ heading; however if it is a type of ASO treatment which is mutation-dependent, it does need mention in the paper, if only to say why it isn’t included.     

Details of a number of more minor points requiring correction are given in the Table below. 

Point

Location

Current

Suggested revision

Comment

3

Section 1.2

Lines 66-80.

The text for each mechanism is currently discussed in the order A,C,B (as labelled in Figure 1).

Since these are out of logical order, the authors can avoid confusion by labelling these with referencing to the particular diagram.

3i

ie. line 68

‘For transcript degradation, ASOs…’ (Figure 1A)….’ 

‘For transcript degradation (Figure 1A), ASOs…’

3ii

Line 73

‘For splice switching ribonuclease-resistant….’ 

‘For splice switching (Figure 1C), ribonuclease-resistant….’

3iii

Line 77-78

‘Alternatively…..  ….to downregulate expression. These approaches…’

‘Alternatively…..  ….to downregulate expression (Figure 1B). These approaches…’

4

Line 78-79

Does ‘These approaches are chosen…’ refer to all 3 approaches, or does it refer to the splice-site skipping approach only ?

If it is all 3 approaches, please indicate as : ‘These three approaches are all chosen……’. If it refers just to approach C, then this seems to be out of order, and should be re-ordered. 

5

Line 79-80

‘such as in the context of… …(DMD)….’ 

If DMD is approach C, then this part of the sentence may also need re-positioning in this paragraph.  

However:- please see Major Comment 2.

 Is the approach to DMD, with exon-skipping to circumvent an aberrant STOP codon, covered by Fig.1A,B or C, ? or does this require a fourth diagram (1D) ?.

6

Line 81

Figure 1A & 1C

Are the ASOs intentionally drawn bridging 2 exons ?

It could be helpful to explain this more in the Legend to Figure 1

7

Line 103

‘Oligo’ 

As the first use of this abbreviation , please also write in full; ie. ‘oligonucleotide (‘oligo’)’

8

Section 2 Cases

2.1  Line 122

‘after a then 6-year-old Mila, …..’

 ‘after a then 6-year-old girl, Mila, …..’

9

2.1  Line 123 

‘diagnosed with Batten disease’

‘Batten disease’ is clinically heterogeneous and usually divided by age-onset in order to give a better clinical picture for all concerned.  It would be best here to refer to this as ‘juvenile-onset’ or ‘late-infantile-onset’ whichever was the term used.

10

2.1  Line 126

‘caused by the accumulation of toxic lipofuscin accumulation’

‘Accumulation’ is duplicated in error in this sentence.

11

2.1  Line 132-3

‘Batten disease’. 

See comment re. line 123.  Which  clinical type of Batten disease is associated with CLN7 mutation ?  Again please specify the clinical age-onset-type of Batten disease.

12

2.1  Line 160

 ‘rapids’ 

‘rapid’

13

2.1  Line 165-6

‘to treat CLN3 mutations’

Which clinical age-oset type of Batten disease has CLN3 mutations ? – please specify.

14

2.2  Line 183-4

‘A-T is managed by treating the present symptoms [43]’

Is any regular screening performed in an affected patient – eg. for cancer ?.  If so, what, and for what type of cancer ?.  If no cancer screening is performed please say so, and perhaps add one sentence as to why not.

15

2.2  Line 200-1

‘…whereby each type of variant may be amenable by a single ASO in different patients…’

This doesn’t quite make sense.  Do the authors mean : ‘…whereby several patients with each type of variant may be amenable to treatment using a single ASO for that variant type…’ ?  - Please rephrase.

16

2.2  Line 203

‘The patient, Ipek….’   

Please specify her sex.  Ie.  ‘The patient, a female infant, Ipek, ….’

17

2.2  Line 204

‘c.8585-13_8598del’  

Is this the correct mutation which was targeted in this patient ?.  The paper in Nature from Kim et al (ref. 39) indicates that specific PCR primers were used which were ‘designed to exclude the non-target c.8585-13_8598del allele’

The targeted disease allele of the treated patient seems to be: c.7865C>T, and the 2nd allele: c.5763-1050A>G.

Please check this and amend accordingly.  Please also specify in words what type of mutation these are (ie. base substitution /aberrant splice site etc.)  If c.8585-13_8598del is still relevant, please specify also that this is a 13bp deletion.

18

2.2 Line 208

‘in c.7865C>T a cell line’

 ‘in a c.7865C>T cell line’   or: ‘in a cell line carrying c.7865C>T’

19

2.2  Line 214

‘and produced restored’

‘and restored’

20

2.2  Lines 202-242. 

Two paragraphs of case history

These paragraphs are too much like story-telling.  It would seem better with a more succinct approach, omitting some of the incidental detail.

21

2.2  Line 237 

‘safety labs’ 

What are ‘safety labs’  - The authors may need a better term here.

22

2.2 Line 238

‘physical exams’

‘physical examinations’

23

2.2  Line 238-9

‘. At the time of publication’

Please say what age the patient was at that point, and what was the duration of treatment which she had to that point.

24

2.3 Line 251-2

‘and was diagnosed with’

Please give here the clinical diagnosis…caused by a……mutation in KCNT1.   Eg.  ‘malignant migrating partial seizures of infancy’    or :  ‘developmental and epileptic encephalopathy’, and with a reference to that diagnosis  eg. this may be Barcia et al. Nature Genet. 44: 1255-1259, 2012.

25

2.3  Line 255-7

‘Previous work unrelated to Valeria had generated gapmer ASOs that were effective at knocking down pathogenic KCNT1 via RNase H degradation of transcript in vivo [47].’

Please specify that this work was in a mouse model

26

2.4  Line 270 - Heading

‘2.4 Locked Nucleic Acids (LNAs) for TNPO2 Knockdown’

Please include the name of the clinical condition in this heading

27

2.4  Line 271-2

‘leading to a diagnosis of a rare TNPO2 single nucleotide variant’

The diagnosis should be stated as a clinical condition (due to the TNPO2 variant);  eg. Intellectual Developmental disorder with hypotonia, impaired speech, and dysmorphic facies (IDDHISD).- and referenced as such (Goodman ID et al,  Am. J. Hum. Genet. 108: 1669-1691, 2021.)

28

3.  Line 295

 ‘also a concern’ 

Why include the word ‘also’ ?, as it dilutes the impact.  Better to say .  ‘Regulatory constraints are an important aspect of development of… ‘  They are there for a very obvious purpose, rather than being a ‘concern’

29

3. Line 295-307. 

Paragraph

Because this paragraph doesn’t go into any detail of regulation, (whereas the title of the paper suggests that it would) the authors need to say here that more detail of each point is given and discussed in the next few paragraphs.

However, the authors should summarise here the criteria for the type of condition that could be considered for N=1 ASO treatment.  Ie. what types of conditions are suitable (eg. fatal, progressive, and with  no other treatment )

30

‘…casimersen, a recently approved oligonucleotide treating Duchenne mus-

cular dystrophy, included three different animal models, studies spanning nearly one year….’,

There may be good reasons why the DMD ASO  ‘casimersen’ took much longer.  Ie. DMD is not so rapidly progressive, and has a quality of life for many years. Also there may be a large number of patients with the same mutation who could benefit, so that any adverse effects are multiplied in terms of numbers.   These should be considered and discussed here, rather than just making the simple comparison.

Comments on the Quality of English Language

Occasional corrections needed.  I have detailed several of these in the Table of more minor points for the authors

Author Response

Reviewer 3: (Comments underlined, responses in blue)

This is a very readable and fascinating paper for general readers, in which the authors point to the future by compiling a current summary of published examples of single-patient antisense oligonucleotide (ASO) treatments for individual rare genetic disorders. The authors also present some of the background considerations regarding the statutory regulation in the US for the development of such treatments, albeit at a quite superficial level, but which is probably inevitable in a summary paper. The narrative style of the case histories, including referring to each patient by name, tends to a flowing journalistic style, rather than the more customary succinct scientific style of writing, and although to some the narrative style could be irritating, it should not necessarily require revision. However, in relation to this there is a Major point that does need attention:

Thank you for taking the time to review our manuscript. Your comments and thoughts are sincerely appreciated.

Major Point

1.Given that each patient’s name is already public, and that the name given to each patient-specific ASO is based on the patient’s name, the use of the patient’s name is understandable. However, the authors do need to add a note in the Acknowledgements and Informed Consent Statement about this, and confirm that the parents of each child has given a general permission for the child’s name to be used in any publication which includes the details of their case. Accordingly, I would question whether the Informed Consent Statement as ‘not applicable’ is an acceptable response by the authors.

Thank you for bringing up this point. We have expanded our informed consent section to mention the public nature of all discussed articles, and the lack of our own human patients in this review study.

The consent segment now reads: “All study and patient information came from publicly available articles and sources which had previously obtained the required consent for each respective case by the original authors. No new patients were included in this review article that had not been previously published.”

The acknowledgements section now reads: “We also wish to thank and commend all the patients and patient families involved in the studies mentioned. Their bravery, selflessness, and commitment have paved the path for many more patients to come.”

  1. The three example-types of ASO in the diagrams in Figure 1, do not make any reference to use of ASOs to give ‘exon-skipping’ as an approach to re-correction of a reading frame in cases where there is an out-of-frame exon deletion. Since this is the most common mutation mechanism in Duchenne muscular dystrophy, the authors need to comment on this, and, if appropriate, either add a fourth mechanism to Figure 1, or mention in the text and legend if it is already covered by one of the three mechanisms present. It maybe, that given the high incidence of DMD, and the number of patients with similar mutations, the authors may not consider this to be within an ‘N-of-1’ heading; however if it is a type of ASO treatment which is mutation-dependent, it does need mention in the paper, if only to say why it isn’t included.

As correctly noted, exon skipping is extremely important due to its prevalence in the context of DMD. Exon skipping is typically classified as a subcategory of splice-switching, which was included in the figure. To improve clarity, this has been more directly mentioned. The section now reads: “For splice-switching, ribonuclease-resistant ASOs are instead designed to bind to important splice sequences such as acceptors, donors, silencers, or enhancers. Rather than stimulating degradation, these ASOs sterically hinder spliceosome binding, leading to the inclusion or exclusion of a given intron or exon [16]. A common type of splice-switching is exon skipping, such as in the context of Duchenne muscular dystrophy (DMD) where the exclusion of mutated exons to restore the reading frame can confer clinical benefit [18].”

Details of a number of more minor points requiring correction are given in the Table below (Table not included due to formatting issues). Table outlines minor grammatical points 3-30

All spelling and grammatical suggestions were incorporated, unless specifically noted below. Areas of major change are also covered below.

Points 3-6: The order of figure subpanels has been adjusted to align with the order that they are discussed in text, with references to each subfigure as appropriate. The figure caption has also been updated to include more information. As per point 2, the clarity of splice-switching/exon skipping has been improved. The section now reads:

“ASO-mediated modulation can occur either through transcript degradation, translational regulation, or splice-switching (Figure 1) [15,16][15,16]. For transcript degradation, ASOs with a sequence complementary to the transcript of interest are designed using base pair and backbone chemistries that are susceptible to ribonuclease H. Upon binding to the tar-get transcript through Watson-Crick base pairing, the RNA-ASO duplex is degraded by ribonuclease, leading to knockdown of the target gene (Figure 1A) [17][17]. This method is ideal when the disease arises from the presence of a pathogenic transcript, such as in toxic gain-of-function mutations. For splice-switching, ribonuclease-resistant ASOs are instead designed to bind to important splice sequences such as acceptors, donors, or enhancers. Rather than stimulating degradation, these ASOs sterically hinder spliceosome binding, leading to the inclusion or exclusion of a given intron or exon (Figure 1B) [16][16]. A common type of splice-switching is exon skipping, such as in the context of Duchenne muscular dystrophy (DMD) where the exclusion of mutated exons to restore the reading frame can confer clinical benefit [18]. Rather than altering splice patterns, ribonucle-ase-resistant Alternatively, the same ASOs can also be targeted to ribosomal binding sites to downregulate expression by impairing ribosomal binding (Figure 1C).”

Point 14: The cancer screening for A-T has been briefly mentioned, and now reads: “. Cancer screening is also an important management tool, and frequent screening is recommended beginning in early childhood, although no definitive guideline exists for cancer surveillance in A-T [44].”

Point 15: This confusing sentence was omitted.

Point 17: Thank you for your attention to detail. The patient is compound heterozygous for both mutations, and it was the c.7865C>T mutation that was targeted with ASOs. This section has been reworded for clarity and now reads: “The female patient, Ipek, was 1 year of age at referral and carried compound heterozygous loss-of-function mutations in ATM; a 13bp deletion (c.8585-13_8598del) in one allele and a c.7865C>T point mutations in the otherin ATM. The latter is predicted to produce a novel splice donor site within exon 53 and cause protein frameshift by excluding 64 bp of the exon in the mature mRNA. Previous work has shown that ASO could effectively block the novel splice site and restore ATM protein function in a c.7865C>T a cell line, providing precedence for N-of-1 ASO treatment for the c.7865C>T allele [46][45].”

Point 20: Due to the limited (often single source) information available for many of the ASOs, and the emphasis on the pathway to N-of-1 ASO development rather than any individual ASO, the authors feel it is important to outline as much of the development pathway as possible, rather than a succinct overview which omits much of the pre-clinical developmental process and rationale.

Point 29: The clarity of the line covering indications for N-of-1 ASO development was improved, and now reads: “N-of-1 ASO development is intended to be used when there are no other FDA-approved options available for a patient, and the disease being treated shows rapid progression leading to death or serious impairment. The guidelines below outline the areas that must be addressed in the investigational new drug (IND) application, with major components including a non-clinical report, chemistry and manufacturing report, and clinical plan.”

Point 30: Major differences certainly exist between the disease manifestations and prevalence of DMD compared to CLN7 Batten disease, however the purpose of this comparison is to highlight the differences between an N-of-1 ASO trial versus a typical clinical trial, rather than an apples-to-apples comparison of two different ASOs. Factors such as slower progression rate and higher prevalence are a justification that N-of-1 trials may not be required, but affect the decision to undertake N-of-1 vs standard trials more than they affect the timeline and requirements of the trial type chosen.

Round 2

Reviewer 2 Report

Comments and Suggestions for Authors

I fully agree with the authors' revision.

Reviewer 3 Report

Comments and Suggestions for Authors

genes-3056569.   Review of N-of-1 ASO therapies.  Revised version.

The authors have satisfactorily responded to my review comments, and incorporated these in this paper.  The authors are to be commended on this now excellent and very readable review which will interest and inform any reader not already familiar with this field.  Well done !